

# 1 Air temperature and light intensity in a tropical rainforest of Brunei
# 2 Darussalam: Time series recorded in 2017

Kazimierz Becek[1,2], Kamaria A. Salim[3]
[1]Faculty of Geoengineering, Mining and Geology, Wroclaw University of Science and Technology, Wroclaw, 50-421, Poland
[2]Department of Geomatics, Zonguldak Bulent Ecevit University, Zonguldak, 67100, Turkey
[3]Department of Biology, University of Brunei Darussalam, BE1410, Brunei Darussalam
*Correspondence to*: Kazimierz Becek (kazimierz.becek@pwr.edu.pl)
**Abstract.** The air temperature and light intensity were recorded in a tropical rainforest of Brunei Darussalam at 20-minute
interval in 2017. HOBO Pendant® data loggers were attached to tree trunks approximately 2 m above ground. The obtained
data can be used to study various microclimatic, biological and ecological characteristics of tropical rainforests in Brunei. The
long term observations can also be used to study the impact of global climate change on the canopy of tropical rainforest. DOI:
10.17632/5vzp6svhwh.3, (Becek 2018).

## 13 1. Background

The goal of this experiment is to identify the impact of global climate change on tropical rainforests (e.g., Bonan 2008). The
forest canopy is considered an interface between the forest's ecosystem and the Earth's atmosphere and appears to be a
promising place to look for the symptoms of this impact (e.g., Nadkarni and Solano 2002). This potential for discovery in the
canopy is one of the reasons that forest canopy studies have been emerging as a field science for some years now (Lowman
2009; Nadkarni et al. 2011). As one of the models of the impact of global climate change on tropical rainforests suggests, the
rising air temperature triggers an increase in the transparency of the forest canopy, which in turn allows for a higher throughfall
of rainfall to the forest floor. Further, this process might lead to the development of a self-sustaining cycle, gradually
destabilising and depleting the tropical forest ecosystem (Becek and Horwath 2017). Some indications of this assertion were
provided by an analysis of the remote sensing satellite data for the Brunei tropical rainforest (Becek and Odihi 2008). A
confirmation of the depletion of the forest canopy's transparency would have a far-reaching impact on the global climate
change perception among contemporary societies as well as on the mitigation strategies for this impact (Becek and Ibrahim
2012). One approach to assessing the forest canopy's transparency and the related changes is to measure the rainfall above the
canopy and the throughfall at select positions distributed underneath the canopy (Baldocchi et al. 1988; Lowman and Moffett
1993; Moffett and Lowman 1995). However, this method is technically much more complicated and costlier to manage due to
the instruments and their maintenance above the canopy. An alternative approach is to use the light intensity data recorded
underneath the forest canopy as a proxy of the transparency or throughfall of the forest canopy. The long-term and very high
temporal resolution air temperature and light intensity time series recorded in an extremely biodiverse environment would also



allow for a number of studies, including ecological, biological, climatological. Such studies could be significantly expanded
and enriched by the inclusion of some other type of data, including the Light Detection and Ranging (LiDAR) datasets (e.g.,
Lovell et al. 2003; Gonzalez et al. 2010) or the remote sensing data from, for example, the Copernicus Sentinel program (ESA
2018a). Since January 2011, twenty HOBO® Pendant data loggers have been recording the air temperature and light intensity
at three locations in a primary rainforest of Brunei Darussalam located on Borneo, just four degrees north of the Equator. The
described dataset of the rainforest's air temperature and light intensity is the first of its kind to be published. More data of the
same type collected at different locations in Brunei Darussalam at different heights above the ground and for significantly
longer periods will be made available in the future.
**2. Data and Methods**
In order to gather the evidence to prove the conjecture that global warming is affecting the transparency of the forest canopy,
i.e., increasing rain throughfall, it was decided to record the air temperature and light intensity at a 20-minute interval for an
extended period, in this case, at least 10 years. It is anticipated that a trend in the air temperature (already confirmed by many
other experiments around the world, e.g., Karl & Knight, 1998) and light intensity will be identified in the time series. For the
stated purpose, the air temperature and light intensity sensors have been installed in locations in the Brunei Darussalam tropical
rainforest. What follows are all the relevant details concerning the test site and a description of the data recording method.
**2.1 Area of Interest**
Figure 1 shows a map, which provides a geographic context for the area of interest (AOI). The approximate coordinates of the
site are Latitude = 4°34'13.5", Longitude = 114°25'06.5". The elevation of the site is approximately 11.1 m above mean sea
level (AMSL). The topography of the AOI is dominated by flat, sandy terraces eroded by rainwater. Small water ponds and
streams with closed watersheds exist in the lowest parts of the landscape. Based on the observations made by the authors, most
of these topographic features host rainwater during monsoon periods; otherwise, the forest floor is moist but not wet. Figure 2
shows a geometrically corrected mosaic of high resolution (10 cm pixel) aerial photographs which were acquired on 18 March
2018. An approximate location of the test site is marked with a green circle.

**Figure 1: Location of the area of interest (AOI) in relation to well-known geographic features. Source: K. Becek – own work.**

Earth System
Science
Data

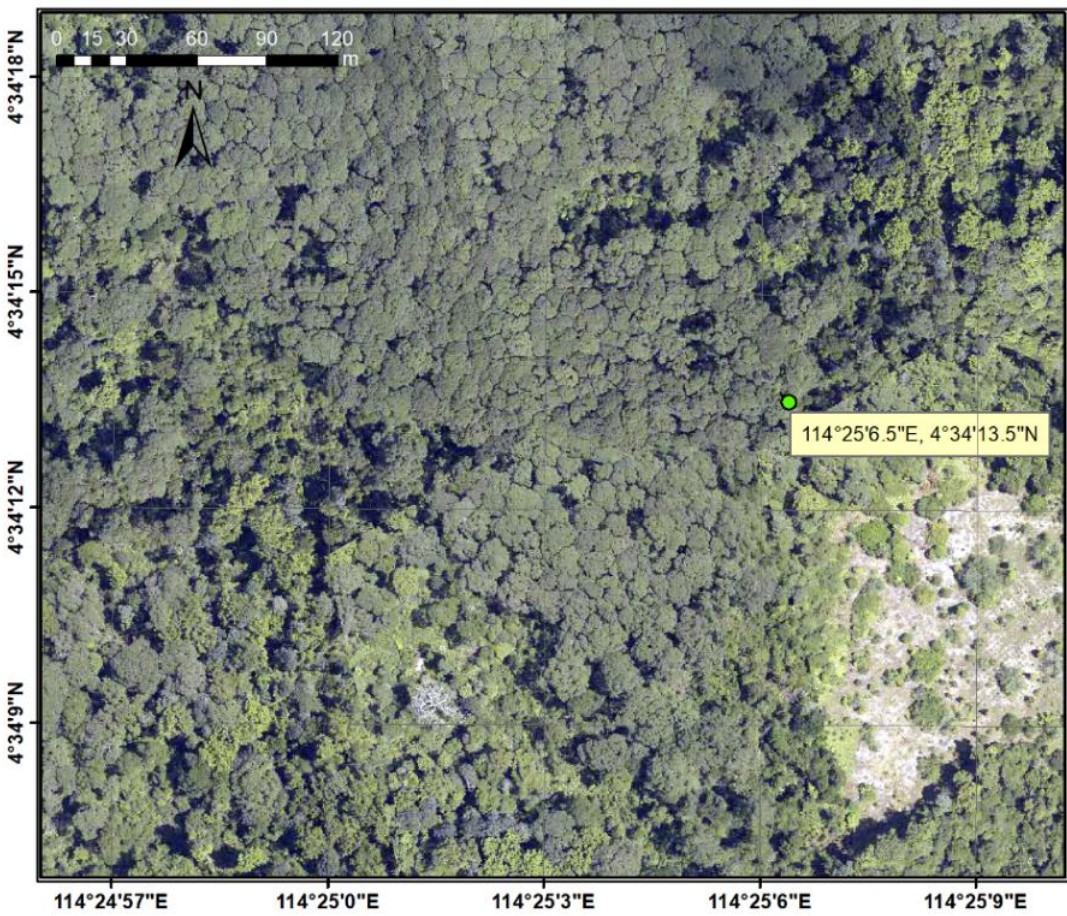


**Figure 2: Geometrically corrected mosaic of aerial images (orthomosaic) taken on 18 March 2018. Source: Courtesy of Soartech Systems Sdn. Bhd., Brunei Darussalam.**

The central part of the AOI is covered with dense forest. In the lower right corner of the images, an area with sparse vegetation on sandy soil is clearly visible. In this area, the forest was destroyed by fire in 1998. A reforestation attempt carried out by local school students has achieved limited success in establishing new forest cover.

Because of the extremely dense and multilayer structure of the forest at the AOI, a high-resolution topography of the site was unknown until a recent high-resolution LiDAR survey was carried out on 18 March 2018. Figure 3 shows a digital terrain model (DTM) of the site derived from the LiDAR points collected on the forest floor at an average density of 0.4 points/m2. The ground pixel size of the DTM is 0.5 m. The prevailing elevation on the DTM is between 11 and 12 m. The slightly brighter spots (elevated spots) in this part of the DTM are buttressed trees roots. Comparing the lowest topography of the DTM with the orthomosaic shown in Figure 2, one can note that the forest is much denser and more spatially coherent species-wise. The higher areas of the DTM clearly correlate with a highly diversified forest in terms of both species and tree height, most likely due to the availability of water and nutrients, which are transported from the elevated areas. This observation corroborates well

with the vicious cycle model outlined in Becek and Horwath (2017). The forest floor of the AOI is extremely uneven, which,
combined with the lowest dense strata of vegetation that includes rattan species, makes the forest barely penetrable to humans.

The prevalent climate at the AOI is described as Tropical Rainforest Climate, according to the Köppen Climate Classification
subtype. The average annual temperature at Brunei International Airport is 27.8°C and on average, the warmest and coolest
months are April (28.3°C) and January (26.7°C). Precipitation in Brunei varies between 2,500–4,500 mmyr-1. The AOI
receives approx. 3200 mmyr-1. For the last forty years, the temperature in Brunei has been rising at the rate of about 0.28°C
per decade, and the monthly mean rainfall has been rising at the rate of 9.8 mm per decade. Dykes (1997, 2000) reported a
valuable research on climate and forest canopy throughfall. The observations fall well within a long-term meteorological data
available from Brunei Darussalam Meteorological Department (BDMD) (although not online). The website of the BDMD
(http://www.bruneiweather.com.bn/climate) provides also a comprehensive and official description of Brunei Darussalam
climate.

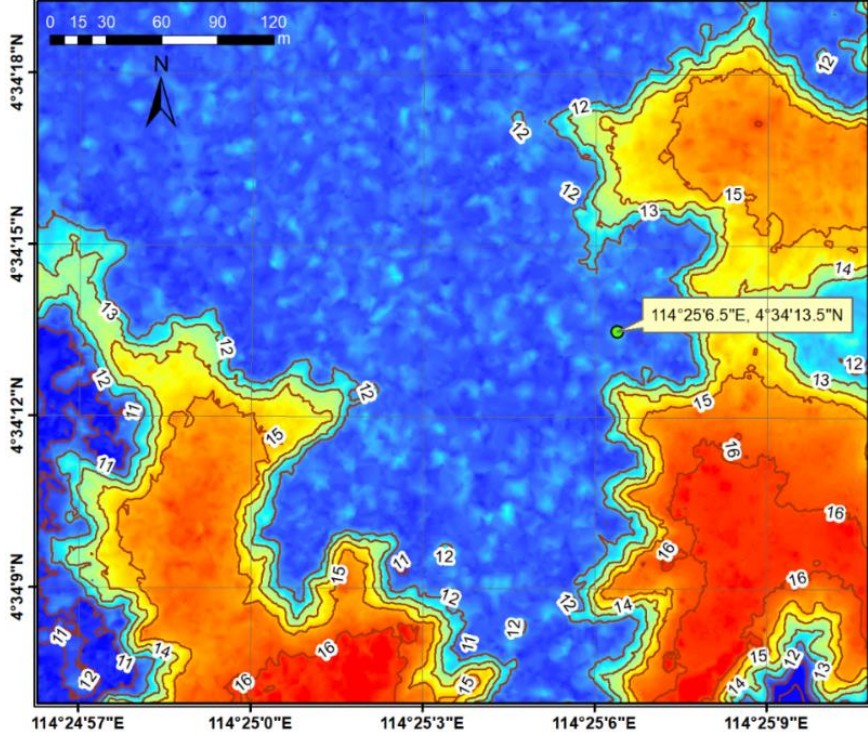


**Figure 3: Digital terrain model of the AOI based on the LIDAR survey. The ground resolution of the model is 0.5 m. The contour**
**interval is 1 m. Source: K. Becek – own work.**
According to the forest map of Brunei Darussalam, the AOI's forest cover is classified as a mixed species, fresh water peat
land forest dominated by Shorea albida Sym. and occasionally by the endangered Agathis borneensis Warb. species.



**Figure 4: Vertical cross-sections N-S (top), and W-E (bottom) over the AOI. The forest strata are as follows: Low < 0.5 m; Medium > 0.5 m but <= 15 m; and High > 15 m. These profiles were produced from the recent LiDAR data. Note horizontal and vertical scale bars. Source: K. Becek – own work.**

A glimpse of the forest canopy from above is seen in Figure 5. This figure was prepared from the LiDAR points captured at an approximate density of 15 points/m$^2$. The radius of the circle is 100 m, and its centre is coincidental with an approximate sensor's position. The canopy trees can be easy recognised, with the tallest located in the lower sections of terrain (compare DTM in Figure 3).

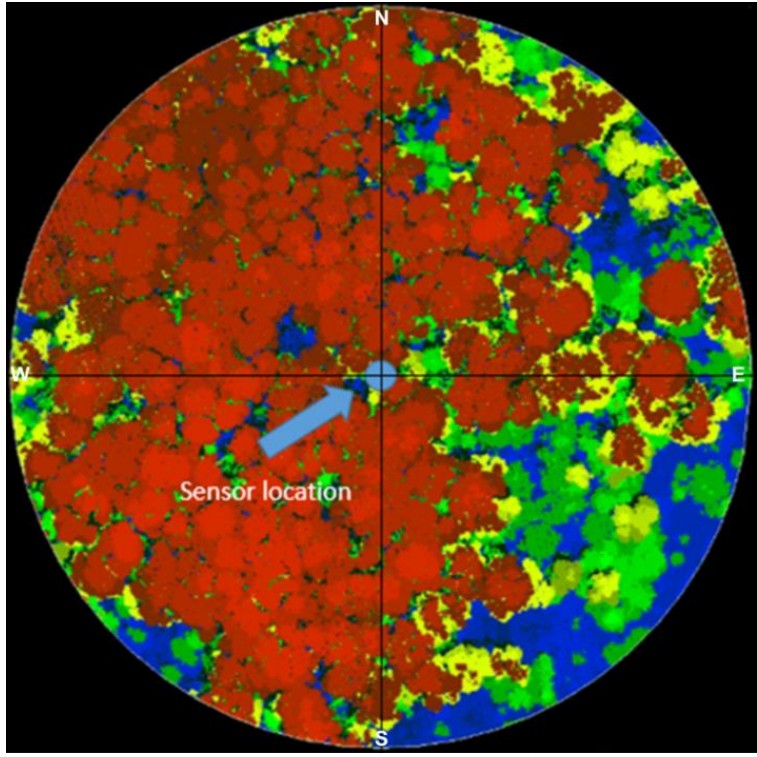

96

**Figure 5: AOI as seen from an above-canopy position. Colours indicate height above the ground: blue –ground surface; light brown < 0.5 m; green > 0.5 m but <= 15 m; and red > 15 m. Source: K. Becek – own work.**

### 2.2 Data

The HOBO® Pendant data loggers were selected as an optimal device for this project. The selected sensor model can store up to 64 kB of the air temperature and light intensity data at selectable temporal resolutions. In this case, a 20-minute resolution was selected as a balance between data storage capacity, frequency of data download requirement, battery life (approximately 1 year) and the temporal gradient of the temperature and light intensity. The 20-minute temporal resolution allows for 400 days of data recording. A detailed description of the HOBO® Pendant logger can be found in the Standard Operating Procedure (SOP) available from the data repository. To maintain a level of redundancy of data recording (e.g., in case of a faulty data logger or mechanical damage to the logger by animals and insects), five loggers were installed no farther than 5 m from each other. The sensors were attached to tree trunks at approximately 2 m above the forest floor using a tree-friendly device developed for the experiment. Figures 6a, b, and c show the HOBO® Pendant data logger, the recorder attached to the deployment device and the data logger mounted on tree trunk, respectively. The loggers were mounted with the sensor facing the forest floor and from the western side of the tree trunks to maximise the diffused irradiance available to the light intensity sensor.



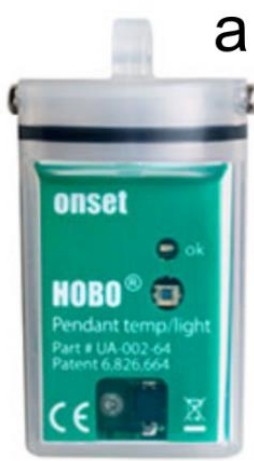
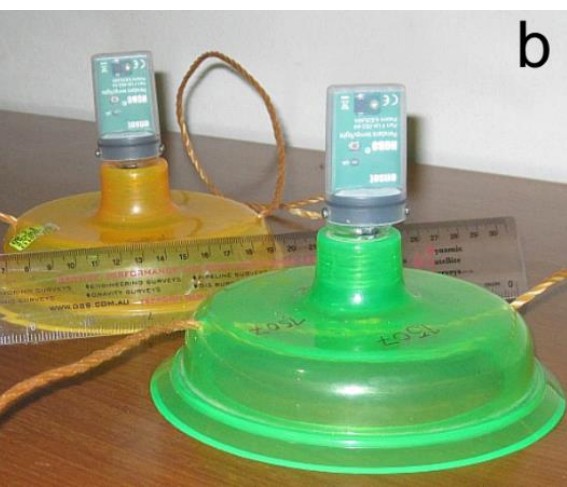
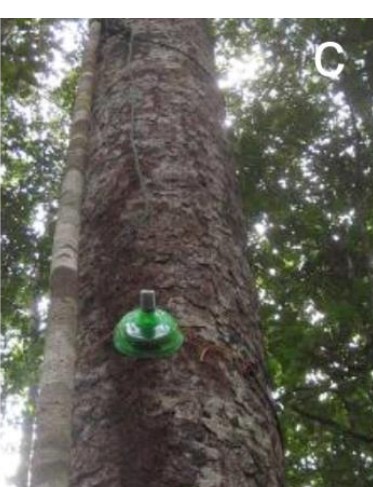


**Figure 6: a – HOBO® Pendant data logger used in the experiment; b – Data logger attached to a deployment device; c – Data logger**
**attached to a tree trunk. Source: K. Becek – own work.**
Three of the five installed loggers produced datasets of the air temperature and light intensity, while two failed to record any
useful data. There are 25,896 records for each recorder. The recording commenced on 6 January 2017 at 6 am local time (+8
GMT) and concluded on 31 December 2017 at noon (+8 GMT). Table 1 shows an example of three records taken from the
logger named 'S1_2017'.

**Table 1: Selected records from the dataset.**

| S/N | Site Code | Sensor | DT (Date/Time) | Temp. (°C) | Intens. (lux) |
|-----|-----------|---------|------------------|------------|---------------|
| 4 | NB | S1_2017 | 06-01-17 07:00:00 | 23.484 | 43.1 |
| 5 | NB | S1_2017 | 06-01-17 07:20:00 | 23.484 | 150.7 |
| 6 | NB | S1_2017 | 06-01-17 07:40:00 | 23.581 | 290.6 |


Figure 7 provides an example of the data graphs for all three loggers for a 24-hour period from 6 am local time (+8 GMT) of
6 January 2017 to 6 pm local time (+8 GMT) of 7 January 2017.

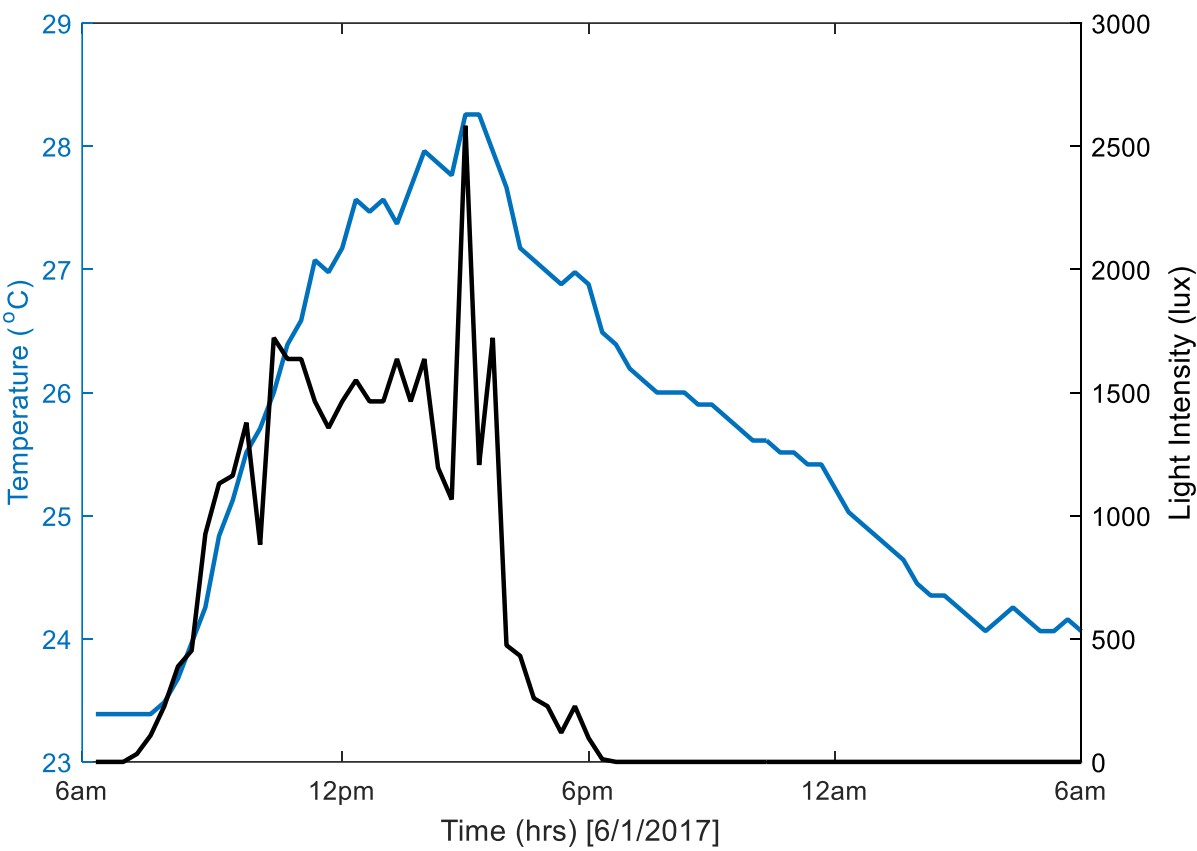


**Figure 7: A sample plot of the temperature and light intensity captured during a 24-hour period from 6 am local time (+8 GMT) of 6 January 2017 to 6 am local time (+8 GMT) of 7 January 2017. Source: K. Becek – own work.**


The repository contains also the LiDAR data and geometrically corrected aerial photography of the AOI. The LiDAR data
records, besides x, y, z coordinates of points, in the first column contain the class of land cover, i.e., 0 and 1 denotes vegetation,
and terrain, respectively. Both data sets are on the WGS84/UTM 50N datum/ map projection. Both datasets were collected on
18 March 2018.
**3. Access to the data and metadata**
The dataset can be accessed using this link: DOI: 10.17632/5vzp6svhwh.3, (Becek 2018). The metadata describing all technical
aspects of the dataset, including the SOP are available under the same link. The current data model is straightforward and
includes two measurable variables of the primary tropical rainforest in Brunei Darussalam, i.e., air temperature and light
intensity captured close to the forest floor. The sole objective for installing more than one logger is to provide redundancy



within the experiment, and the data from these additional loggers can be used as a control. Table 2 shows some basic statistics
of the differences between the corresponding records of three loggers.

**Table 2: Basic statistics of the differences in the temperature and light intensity between sensors (S1, S2 and S3).**

| Sensors' Pair | Mean Temp Diff. (°C) | RMSE Temp. Diff. (°C) | Mean Intens. Diff. (lux) | RMSE Intens. Diff. (lux) |
|---|---|---|---|---|
| S1-S2 | 0.0067 | 0.372 | 59.2 | 514.5 |
| S2-S3 | 0.0348 | 0.339 | -139.4 | 569.5 |
| S1-S3 | -0.0415 | 0.352 | 80.2 | 526.8 |


It was found that the root mean square error (RMSE) of the differences in temperature is within the manufacturer's specified
limit (RMSE = 0.53°C > 0.372°C). In addition, it appears that there is no temperature bias present in the recorded data (Mean
temperature difference range is from -0.0415°C to 0.0348°C). The light intensity readings are approximately 30% of the
average daytime light differences (~1400 lux). This apparently large can be justified by two possibilities. First, the loggers are
located at various locations no more than 5 m from each other. However, the spatial variability of the light intensity at the
bottom of the forest is high, which leads to a higher data difference between loggers. Second, according to the manufacturer
of this logger model, the logger was 'designed for measurement of relative light levels' and not absolute light intensity levels.
A careful study of the first and last light times confirms that the first possibility, i.e., the location of the loggers, appears to be
responsible for the high differences between the loggers' data. This information could be used to further study, for example, a
spatial correlation between the three records.
**4. Data availability**
The data sets, metadata and the standard operating procedure described in this contribution are available from the Mendeley
Data repository (Becek, K.: Air temperature and light intensity in tropical rainforest of Brunei Darussalam in 2017, Mendeley
Data, DOI: 10.17632/5vzp6svhwh.3, 2018).
**5. Conclusions**
The spatially and temporally coincidental air temperature and light intensity data, even without meteorological data from
outside the forest, provides unique material to study various ecological, microclimatic, biological, thermodynamic,
phenological (Morisette et al 2009) and other characteristics of a primary tropical rainforest of Brunei Darussalam and beyond.





The dataset from this research was recorded at a very high temporal resolution and for an extended period (nearly one year).
Interannual datasets will be made available to the open repository as they become available. Five sensors deployed at the test
site provide a highly redundant, robust and reliable dataset. The light intensity, for example, studied together with the sun's
position in the sky (geographic coordinates of the sensors provided), can supply insight into diurnal and seasonal information
regarding the variability of the light conditions on the forest floor (Hubbel et al 1999; Yamamoto et al 2011), the transparency
changes of the forest canopy, internal structure of the forest canopy and others. This information can be used in connection
with remote sensing data captured by, for example, the Copernicus Sentinel program, and in particular, with the Sentinel 1A/B
and Sentinel 2A/B (currently in orbit) missions' data (ESA 2018a). In addition, some very sophisticated missions are being
prepared or are already in orbit, including IceSAT-2 data (NASA, 2018) and the Biomass Mission (ESA 2018b). A current
limitation of the dataset is the lack of corresponding data loggers outside of the forest, which would provide a useful and
powerful reference dataset to expand possible study options. Another limiting factor of the experiment is the lack of real-time
sensor monitoring. However, improvements in Brunei's mobile phone coverage has been improving and soon will provide the
technical means to overcome some of these communication limitations.

It is planned to add datasets to the repository as they become available. These datasets will include the following:

1. Data sets for additional two locations in different forest stands.
2. Data sets from loggers fixed at various heights above the forest floor.
3. Datasets collected after January 2011.
**Author contribution**
KB designed the experiment, provided initial funding, installed sensors, download the data and carried out it's maintenance.
KS helped with obtaining relevant permissions and financially contributed the experiment. KB prepared the manuscript with
contributions from KS.
**Competing interests**
The authors declare that they have no conflict of interest.

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
