# Peer review of "Air temperature and light intensity in a tropical rainforest of Brunei Darussalam: Time series recorded in 2017"

_Earth System Science Data, 2019_

## Referee Comment (RC1) · Anonymous Referee #1 · 20 Aug 2019

The authors provide unique air temperature and light intensity data for a site in a tropical rainforest of Brunei Darussalam. The data comprises time series recorded in 2017. Furthermore, the authors intent to supplement the repository with further time series starting from January 2011. The experiment is planned for ten years period. So long term observations can be used to study various issues related to global climate change and its impact on tropical rainforest. Moreover, the data is unique due to sensors location. The text seems to be well-written and easy to understand. Data acquisition is well documented. However, Figure 4 should be addressed in the text if it is provided. Besides the mentioned times series also LiDAR data and orthomosaic as auxiliary data are provided for the site of interest. However, there are two concerns related to the

auxiliary data: 1. Likely, something went wrong while saving the orthomosaic. Please check it and save this file as a GeoTIFF file if necessary. Anyway, I had problems to read this file with preserving georeferencing. 2. LAS format is meanwhile commonly accepted format for LiDAR data exchange. Please provide the point cloud in LAS format. This data in the txt-format is less useful.

---

## Referee Comment (RC2) · Anonymous Referee #2 · 12 Sep 2019

This is a very interesting proposal that seems likely to collect some useful data. However, it reads like a research proposal rather than a reporting of work that has been done. In addition there are some key aspects the really need addressing.

Much of the work is "anticipated". It seems like a great plan, it just seems to belong in a proposal rather than a journal article.

The first that needs to be addressed with the correlation between the fall through of rain and the light transmittance. If one is going to be used as a proxy for the other, it would be nice to have some idea of the relationship between them. I suspect that it is monotone, but also nonlinear. Understanding this relationship - in at least one location

in the project zone - would enable a better metric for not just explaining that the canopy is becoming more transparent, but how much that would affect the fall through.

Finally, there is one sentence that seems to be mis-worded. Line 23 on the first page "depletion of the forest canopy's transparency" would imply that the transparency is decreasing - making the canopy more opaque. I think that they mean that the depletion of the forest canopy is causing -increased- transparency.

---

## Author Comment (AC1) · 2 Nov 2019

Comment 1: The authors provide unique air temperature and light intensity data for a site in a tropical rainforest of Brunei Darussalam. The data comprises time series recorded in 2017. Furthermore, the authors intent to supplement the repository with further time series starting from January 2011. The experiment is planned for ten years period. So long term observations can be used to study various issues related to global climate change and its impact on tropical rainforest. Moreover, the data is unique due to sensors location. The text seems to be well-written and easy to understand. Data acquisition is well documented.

A1: Thank you for this comprehensive summary of the manuscript.

————-

Comment 2: However, Figure 4 should be addressed in the text if it is provided.

A2: This issue will be addressed in the next version of the manuscript.

————-

Comment 3: Besides the mentioned times series also LiDAR data and orthomosaic as auxiliary data are provided for the site of interest. However, there are two concerns related to the auxiliary data: 1. Likely, something went wrong while saving the orthomosaic. Please check it and save this file as a GeoTIFF file if necessary. Anyway, I had problems to read this file with preserving georeferencing.

A3: The new version of the datasets, including the orthomosaic in question, were uploaded to the repository: DOI: 10.17632/5vzp6svhwh.4. https://data.mendeley.com/datasets/5vzp6svhwh/4 The orthomosaic is in GeoTIFF format (WGS84/UTM50N).

————-

Comment 4: LAS format is meanwhile commonly accepted format for LiDAR data exchange. Please provide the point cloud in LAS format. This data in the txt-format is less useful.

A4: The latest version of the datasets contains the LiDAR data in the las v 1.2 format.

---

## Author Comment (AC2) · 2 Nov 2019

Comment 1: This is a very interesting proposal that seems likely to collect some useful data. However, it reads like a research proposal rather than a reporting of work that has been done.

A1: This manuscript is designed to comprehensively describe all technical details concerning the long-term experiment, of which the first phase is the acquisition of time series of air and light intensity data over a long time period. Naturally, our intention is to use the data to investigate the issue indicated in the manuscript, i.e., the interaction between meteorological/climate variation and transparency of the forest canopy. We

deliberately did not mention what kinds of possible research projects can be derived from the datasets; rather, we hope that publication of the one-year data will stimulate other researchers to develop their own ideas about possible applications in their relevant disciplines.

————————

Comment 2: In addition there are some key aspects the really need addressing. Much of the work is "anticipated". It seems like a great plan, it just seems to belong in a proposal rather than a journal article. The first that needs to be addressed with the correlation between the fall through of rain and the light transmittance. If one is going to be used as a proxy for the other, it would be nice to have some idea of the relationship between them. I suspect that it is monotone, but also nonlinear. Understanding this relationship - in at least one location in the project zone - would enable a better metric for not just explaining that the canopy is becoming more transparent, but how much that would affect the fall through.

A2: The purpose of this paper is not to show or investigate the relationship between the fall through of rain and the light intensity at the bottom of a forest, rather to describe the setting for the experiment and the data that have been captured by the sensors. This is why we don't discuss or show any results concerning the said relationship or correlation. Our future papers, published somewhere else, and not in this particular journal (which is designed to promote and describe environmental data relevant for wider audience) will be discussing this correlation between the light, temperature and the fall through of rain, and possibly other issues.

————————

Comment 3: Finally, there is one sentence that seems to be mis-worded. Line 23 on the first page "depletion of the forest canopy's transparency" would imply that the transparency is decreasing - making the canopy more opaque. I think that they mean that the depletion of the forest canopy is causing -increased- transparency.

A 3: You are correct. In the new version of the manuscript this will be rectified.